# Socioecological drivers of mutualistic and antagonistic plant-insect interactions and interaction outcomes in suburban landscapes

**Gordon Fitch** [1,2,3]\*, **Lynn S. Adler**[3], **Rebecca E. Irwin**[4], **Paige S. Warren**[5]

**1** Centre for Bee Ecology, Evolution and Conservation, York University, Canada, **2** Dept of Biology, York University, Canada, **3** Dept of Biology, University of Massachusetts, Amherst, MA, United States of America, **4** Dept of Applied Ecology, North Carolina State University, United States of America, **5** Dept of Environmental Conservation, University of Massachusetts, Amherst, MA, United States of America

\* gmfitch@yorku.ca

**Data Availability Statement:** All data and code associated with this manuscript are available at https://osf.io/fj45b/ (DOI 10.17605/OSF.IO/FJ45B).

## Abstract

Cities are complex socioecological systems, yet most urban ecology research does not include the influence of social processes on ecological outcomes. Much of the research that does address social processes focuses primarily on their effects on biotic community composition, with less attention paid to how social processes affect species interactions. Linking social processes to ecological outcomes is complicated by high spatial heterogeneity in cities and the potential for scale mismatch between social and ecological processes, and the indicators used to assess those processes. Here, we assessed how social and ecological processes jointly influence the frequency and outcomes of species interactions among the native perennial vine *Gelsemium sempervirens* and its insect pollinators, nectar robbers, and florivores across 28 residential subdivisions in the Research Triangle region, NC, USA. We integrated data on socioeconomic attributes (mean property value, mean property size, subdivision age), vegetation attributes (forest cover and richness and density of managed and unmanaged floral resources), species interactions (conspecific and heterospecific pollen deposition, nectar robbing, florivory), and *Gelsemium* reproduction (fruit set, seeds per fruit) using structural equation modeling to understand the causal links between socioeconomic attributes, vegetation attributes, and interaction frequency and outcome. Among socioeconomic attributes, property value was the strongest predictor of interaction frequency, having both direct and vegetation-mediated indirect effects on pollination and florivory. However, the effect of socioeconomic attributes on plant reproduction was small. Overall, we were able to explain only a small amount of the variation in any species interaction or reproduction measure. This may be due to the functional similarity of subdivisions, despite large variation in both socioeconomic and vegetation attributes, or may reflect scale mismatch between the ecological and socioeconomic variables. Our findings highlight the need to develop scale-appropriate indicators to improve our understanding of the links between social and ecological processes in urban landscapes.

**Funding:** This work was funded by the US National Science Foundation (NSF-DEB500 0742923 to LSA and DBI-2109520 to GF). The funder had no role in study design, data collection and analysis, decision to publish, or preparation of the manuscript. Any opinions, findings, and conclusions or recommendations expressed in this material are those of the author(s) and do not necessarily reflect the views of the National Science Foundation.

**Competing interests:** The authors have declared that no competing interests exist.

## Introduction

Urbanization is a major driver of environmental change; the global extent of urban land cover increased from 350,000 km$^2$ in 1992 to 740,000 km$^2$ in 2015 [1], and the rate of conversion is expected to increase in the coming decades [2]. Urbanization has profound effects on ecosystem function and biotic communities; cities are characterized by, among other features, habitat fragmentation, biotic homogenization, and high productivity [3, 4]. As a result of these changes, urban environments support biological communities that are often distinct from adjacent, non-urban counterparts [5, 6].

While the bulk of research on the effects of urbanization on biotic communities has focused on changes to species composition, urbanization also has the potential to affect species interactions and the structure of interaction networks [7]. Indeed, species interactions may be particularly sensitive to environmental change, leading to altered interaction patterns even in the absence of changes to species composition [8–11]. Thus, changes to interaction patterns may presage species loss and can act as an early warning system of more permanent and dramatic ecological change. Moreover, species interactions underpin multiple ecosystem services, including pollination and biological control, making it important to understand how environmental changes such as urbanization influence species interactions. Despite this, our understanding of the ways urbanization influence species interactions is in its infancy, and the available research suggests a high degree of context dependence, precluding generalization [7]. This heterogeneity in response to urbanization points to the complexity of urban ecosystems and the multiple mechanisms by which urbanization-associated environmental change may influence species interactions (e.g., top-down vs. bottom-up effects).

Cities are complex socioecological systems, in which social and ecological processes profoundly influence one another [12–14]. Understanding the drivers of ecological patterns in urban ecosystems therefore requires explicit consideration of social processes and patterns, in addition to strictly ecological drivers. Yet multiple characteristics of urban ecosystems complicate the task of understanding how social and ecological processes shape biological communities and patterns of species interaction. First, urbanization influences many environmental parameters simultaneously [15]. Second, urban environments are characterized by high spatial heterogeneity over small scales, yet measures traditionally used in ecological research to assess habitat heterogeneity often fail to capture this complexity [16, 17]. Third, the scales at which social processes operate to shape the urban environment are likely to be different from the scales at which organisms perceive and interact with their environment [18, 19]. One approach to increase tractability is to compare biotic communities or interactions in a single habitat type, such as parks or community gardens, across the urban landscape, leveraging the relative similarity of the focal habitat across sites to gain understanding of landscape composition effects on biotic communities (e.g., [20, 21]). Here we propose and evaluate another approach, taking advantage of the reduced within-development heterogeneity of master-planned residential developments, or subdivisions.

Subdivisions are a common feature of suburban land use in North America, and comprise houses built in the same era, often by a single developer, leading to relative homogeneity in both the built environment and landscaping. Developers' landscaping choices have long-term legacy effects that can persist for decades [22, 23]. Moreover, homeowner associations (HOAs) or similar entities commonly influence landscaping decisions and vegetation management in subdivisions, constraining variation in vegetation structure and composition [24, 25]. As such, subdivisions are likely to be more homogeneous in both socioeconomic and environmental conditions when compared with non-master-planned urban neighborhoods. Therefore, subdivisions may provide a 'natural experiment' to study effects of specific attributes of urbanization

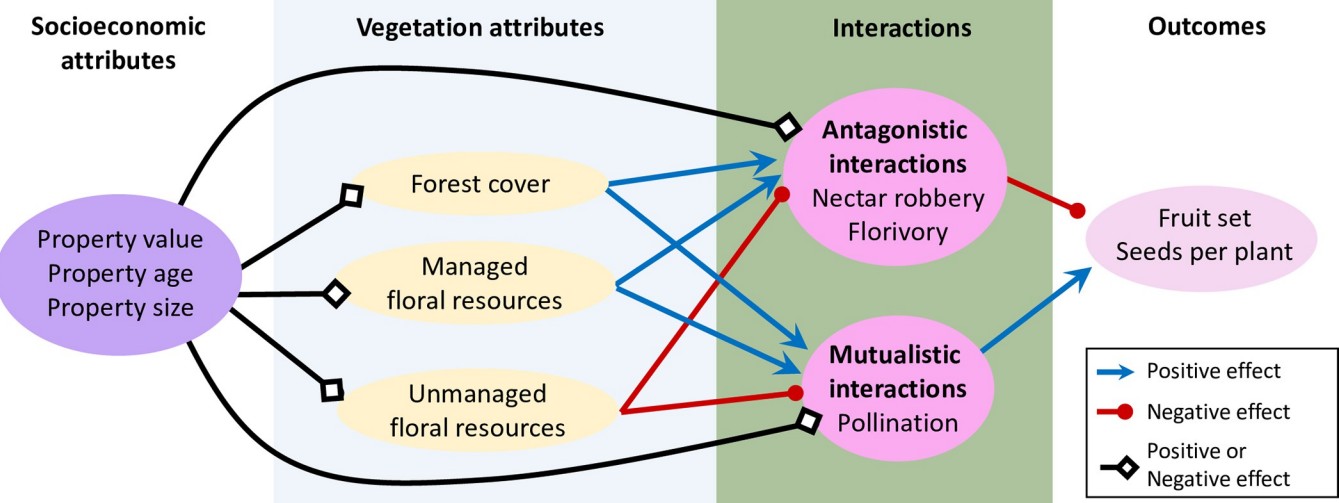

**Fig 1. Hypothesized pathways linking socioeconomic attributes, vegetation attributes, *Gelsemium*-insect interactions, and outcomes for *Gelsemium* reproduction in suburban landscapes.**

on biotic communities, with each subdivision representing a 'patch' with a distinct socioecological profile. Comparing ecological patterns or processes among subdivisions that occupy a common landscape could allow for the identification of causal pathways linking social processes to ecological outcomes.

Here, we leverage a network of subdivisions in the rapidly urbanizing Research Triangle region of North Carolina, USA, to evaluate the degree to which socioeconomic factors structure biotic interactions between the native perennial vine *Gelsemium sempervirens* (hereafter, *Gelsemium*) and its insect visitors. Specifically, we asked whether socioeconomic attributes of subdivisions were linked to interaction frequency and plant reproduction either directly or indirectly via effects on vegetation characteristics that in turn influenced *Gelsemium* interactions. We used piecewise structural equation modeling to investigate the links among socioeconomic attributes, vegetation attributes, and the frequency and plant reproductive outcome of antagonistic and mutualistic plant-insect interactions. As proxies for antagonistic interactions, we assessed the proportion of flowers with evidence of nectar robbery and/or florivory, and the proportion of fruits with herbivore damage; for mutualistic interactions, we used pollen receipt as a proxy for pollination. We hypothesized that socioeconomic attributes would influence interactions both directly (e.g., by influencing management intensity [26]) and indirectly via their effects on vegetation attributes [27, 28], though we did not have *a priori* expectations for the direction of these effects (Fig 1). All the insect species involved in the interactions we assessed are dietary generalists, so we predicted that vegetation attributes such as high floral density and richness would increase populations of antagonists and pollinators, and would therefore increase interaction frequency [29, 30]. At the same time, greater floral resources can decrease interaction frequency via a dilution effect [31, 32]. We predicted that dilution would dominate at local scales (i.e., in the 20m radius around focal plants), leading to negative correlations between unmanaged floral resources and interaction metrics. But we predicted that the net effect of landscape-scale floral resource availability would be increased interaction frequency, leading to a positive association between managed floral resources– which represented the majority of total floral resources in the landscape–and interaction metrics (Fig 1). We expected conspecific and heterospecific pollen receipt to be strongly correlated, but predicted that increased non-*Gelsemium* floral density and richness would lead to

more pollinator switching and therefore increase heterospecific pollen receipt [33, 34]. We expected socioeconomic and vegetation attributes to influence reproductive outcomes for *Gelsemium* only via effects on the frequency of species interactions (rather than direct effects), and that increased interactions with antagonists and heterospecific pollen would reduce reproductive success whereas increased conspecific pollen receipt would increase *Gelsemium* reproduction, given previous evidence for pollen limitation [35]. By integrating social and ecological drivers of plant-insect interactions into a single causal network, this work increases mechanistic understanding of the influence of urban development on species interactions.

## Methods

### Study system

*Gelsemium sempervirens* (Loganiaceae) is a perennial vine native to the southeastern US. It grows in the forest understory and edge, and flowers in the early spring (Mar-Apr). Flowers are large and showy, and are attractive to many insects, including mutualists (generalist pollinating bees) and antagonists (nectar robbing carpenter bees, *Xylocopa virginica* (Apidae), and the florivore *Amphipyra pyramidoides* (Noctuidae)) [36–38]. *Gelsemium* is self-incompatible, and reproduction is often pollen limited, with populations in urban and suburban landscapes experiencing more pollen limitation than those in intact forest [35, 39]. *Gelsemium* is distylous, with plants bearing either pin- (long styles and short stamens) or thrum-morph (short styles and long stamens) flowers [40]. Flower morph influences interactions between *Gelsemium* and both florivores and pollinators, with thrum plants receiving less pollen and florivores tending to consume the more exerted floral organ [37, 41].

### Data collection

**Site selection.** The Research Triangle region of North Carolina has experienced rapid population growth in the past 50 years, and suburban development is extensive. These areas are characterized by extensive developments of primarily single-family homes at moderate densities (>2.5 buildings ha$^{-1}$). *Gelsemium* is common in remnant forest fragments in suburban areas. *Gelsemium* is also grown horticulturally, and so it is possible that some of the wild-growing plants we studied originated from seeds or pollen dispersed from horticultural plantings. To identify potential sites, we extracted a list of subdivisions in Raleigh (N = 60; www.city-data.com). We screened sites for forested patches using Google aerial imagery, and then searched all neighborhoods containing forested patches for *Gelsemium*. This yielded a subset of 28 subdivisions with forest patches containing wild-growing *Gelsemium*; to this list, we added 8 additional subdivisions where we had studied *Gelsemium* previously [42]. From the 35 potential subdivisions, we excluded 8 because they had fewer than 5 flowering *Gelsemium* and/or we were unable to get permission to carry out the research. This yielded 28 subdivisions where we collected field data (S1 Table).

In spring 2010, at each subdivision we selected and marked up to 21 budding *Gelsemium* growing in forest fragments within the subdivision (minimum of 6 plants; mean 16 plants per subdivision). We selected plants that could be distinguished as separate individuals and had at least 3 flowering buds within reach using a 1.2 m ladder. We measured floral traits and fruit and seed set from all plants, and randomly selected 3 focal plants per subdivision to conduct 10 m radius floral surveys, census species interactions, and measure reproduction.

**Socioeconomic and land cover attributes–data sources.** We characterized the socioeconomic attributes of the subdivisions using parcel level information from the County tax assessors' databases (Wake County for most subdivisions, with some falling in Chatham and Durham counties), extracted in 2010. We defined the boundaries for the subdivisions using

GIS layers from the Raleigh iMaps FTP portal for sites in Wake County and from the Chatham and Durham County GIS portals for sites in those counties. For all parcels within a subdivision's boundary, we extracted the parcel's assessed value (property value), the year the home was built (property age), and the size of the parcel in acres (property size). In calculating mean values for the three variables, we excluded all parcels that were not residential, such as sewer/water, HOA lands, historic sites, commercial sites, and we excluded all parcels with a value listed as $0. Population density data was not available at the subdivision level for all sites so was not included in analyses. For subdivisions with data available, population densities ranged from 23–1409 people km$^{-2}$. Property value is an indicator of wealth and socioeconomic status, which we predicted would be positively associated with the density and diversity of managed floral resources, a pattern observed in other North American urban areas [28, 43]. Property age is related to the types of yard plantings and the amount of time they have been in place–both of which can be related to the availability of managed floral resources [43, 44]. Lot size is an indicator of the amount of plantable yard space, and therefore the availability of managed floral resources. We calculated the percentage of forest land cover (forest cover) for each subdivision, using land cover layers from the National Land Cover Database (but see S1 Table). We predicted that forest cover would be related to socioeconomic attributes of subdivisions because choices made during the development process (e.g., house size, lot size) would simultaneously influence property value and remaining forest cover, while age since development would be related to post-development forest regrowth.

Socioeconomic attributes (property value, property age, and lot size) varied substantially among subdivisions, and were uncorrelated with one another; forest cover was positively correlated with age but uncorrelated with other socioeconomic attributes (S1 and S2 Figs).

**Floral surveys—Transects.** We quantified managed floral resources by conducting two, 100 m roadside transects in each subdivision along randomly selected streets. An observer walked each side of the street, identifying all the flowering plants in each front yard in the transect. The width of each transect was defined as the distance from the street to the front of the house, in order to capture a representative sample of plants. For example, most homes had plants along the front of the house ('foundation plantings') that a standardized transect width from the edge of the street might miss in subdivisions with large setbacks. Thus, all data on floral resources were calculated as a density to correct for this variable width of the transect.

We identified all flowering plants to the finest taxonomic level possible, which in most cases was to the species level. When observers could not identify a plant, we took photographs for later identification. We counted the number of individual plants of each morphospecies and estimated the total number of open flowers per morphospecies for at least 3 plants per yard (the largest, the smallest, and one medium sized). We then calculated floral density for each plant morphospecies as the product of the number of plants by the average number of open flowers. For small lawn plants like clover or chickweed, we estimated the area covered by these plants and from that made an estimate of their floral density. We summed all these estimates for each transect to calculate total floral density, and we calculated total floral richness as the number of morphospecies per transect.

**Floral surveys—Radius.** We conducted floral surveys twice on 2–3 focal plants per subdivision between April 9 and 23 (but see S1 Table). Within a 10 m radius around each focal plant, we counted and identified all flowering plants to species, excluding any that appeared to be primarily wind-pollinated (e.g., pine trees and grasses). We also recorded the number of open flowers on each surrounding plant. During the second census we also estimated the percent canopy cover at above each focal plant and 5 m away in a randomly chosen direction using a densiometer. Floral density and richness were calculated as for transect surveys, above.

**Floral traits.** We measured floral traits on all plants once per subdivision between April 9 and 22. We recorded floral morph (pin or thrum), and haphazardly selected 3 flowers per plant to measure corolla length (from tube base to petal flare), corolla width (diameter at the widest point of the corolla tube), petal length (from where petals separated to petal tip) and petal width (across the widest point where petals were separate from the tube) using digital calipers. We used principal component analysis (PCA), with the four morphology traits as dependent variables, to extract morphology measure(s) to include in our SEM. Results from the PCA indicated that PC1 explained 66.7% of the variance, and all four measures were most closely, positively correlated with it; PC2 explained only an additional 14.9% of the variance. All variables had positive loading on PC1; loadings on PC2 were low and a mix of positive and negative. As such, we interpreted PC1 to correlate with flower size, and used it as our measure of *Gelsemium* flower size.

**Censusing *Gelsemium* interactions.** We censused *Gelsemium* interactions on the same days we conducted radial floral surveys around each plant. For each focal plant we estimated plant size (which may be correlated with age) by measuring stem diameter in mm at the plant base, and counted the number of open flowers and the number we could census (on some plants we could not reach all flowers because they were too high in the canopy). We counted the number of flowers with florivory. The carpenter bee *Xylocopa virginica* makes very distinctive slits at the corolla base [37], and we counted the number of flowers with one, two, three, or four or more nectar robbing slits. Because robbing was rare in the study year (<10% of plants experienced robbing) we categorized robbing as yes/no for each plant.

We used pollen deposition per stigma to estimate pollination. Although ideally, we would first emasculate flowers by removing undehisced anthers to prevent self-pollen deposition, this was not feasible with the large number of sites in this study. We therefore collected up to three stigmas per plant from un-emasculated flowers on all plants in the study on the same dates that we censused interactions. Because flowers are distylous, the spatial separation between anthers and stigmas should have reduced, albeit not eliminated, self-pollen transfer. We collected stigmas that were still fresh (not brown or dried) after the corollas had senesced and fallen off. We mounted the stigmas on slides using fuchsin jelly [45] the same day they were collected, and counted all *Gelsemium* and non-*Gelsemium* pollen (excluding pine pollen, which is wind dispersed).

**Fruit and seed set.** We marked the stem beneath every flower we could reach on each focal plant with tape and collected developing fruits approximately weekly from July 14 through August 11, noting for each fruit whether there was evidence of fruit herbivory. We then dissected fruits to count the number of developing seeds and aborted ovules. *Gelsemium* fruits take six months to fully mature and dehisce; we collected fruits while they were still green but large enough to distinguish developing and aborted seeds, but we could not measure mature seed weight. We calculated the ratio of collected fruits to marked flowers to estimate proportion fruit set per plant, and calculated the mean seeds per fruit for each plant. As a measure of fruit herbivory intensity, for each plant we calculated the proportion of collected fruit where herbivory was observed. Signs of fruit herbivory were typically symmetrical holes in the fruit, sometimes with larva, frass and/or seed damage when the fruit was opened. Where any of these signs were present, we recorded 'yes' for fruit herbivory.

## Data analysis

All analyses were conducted using R v4.2.3 [46].

To explore whether subdivisions differed in managed floral resources, we ran negative-binomial generalized linear mixed-effects models (GLMMs) for both floral richness and

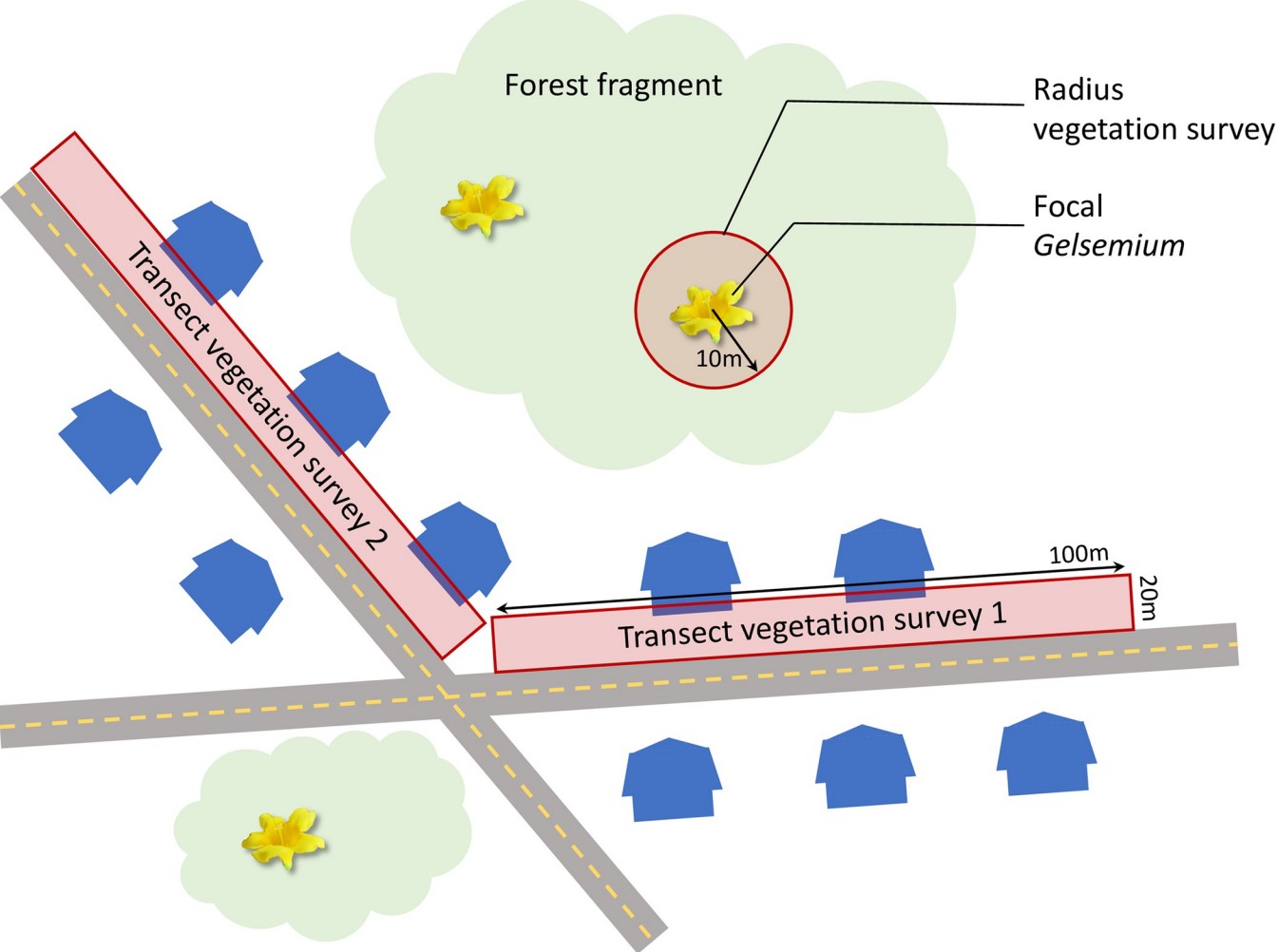

**Fig 2. Schematic diagram of spatial arrangement of study sites.** One or more forest fragments containing focal *Gelsemium* were embedded in a matrix of residential development. Focal plants were censused for evidence of florivory, nectar robbing, and pollination (estimated as pollen deposition), and were assessed for fruit and seed production. We conducted floral surveys at two scales: 10 m-radius surveys around a subset of focal plants (local-scale unmanaged floral resources) and 20 m x 100 m-transect surveys along roadsides within the subdivision (landscape-scale managed floral resources).

density, with subdivision as a fixed effect and property as a random effect. To assess the degree of within-subdivision heterogeneity in floral resource availability, we compared the difference in subdivision-level mean floral richness and density between the most and least flower-rich subdivisions to the range of floral richness and density among properties (for managed floral resources) or radii (for unmanaged floral resources) within a subdivision.

We conducted confirmatory path analysis using piecewise structural equation modeling (SEM), implemented with the 'piecewiseSEM' package [47], to identify relationships among subdivision socioeconomic attributes, vegetation attributes, species interactions, and *Gelsemium* reproduction (Fig 2). We defined component submodels for the following response variables: forest cover, unmanaged floral density, unmanaged floral species richness, managed floral density, and managed floral species richness (vegetation attributes); probability of nectar robbery (yes/no), number of flowers damaged by florivores, *Gelsemium* pollen load, and non-*Gelsemium* pollen load (interactions); and fruit set and seeds produced per fruit (plant reproduction outcomes; see Table 1 for details of model formulation). We used number, rather than

**Table 1. Component model formulations for SEM analysis.**

| Response | Model type | Error distribution | Fixed effects | Random effect |
|---|---|---|---|---|
| **Forest cover** | Linear | Gaussian | Property value<br>Property age<br>Property size | *NA* |
| **Unmanaged floral density** | Linear mixed effects (LMM) | Gaussian | Property value<br>Property age<br>Property size<br>Forest cover | Subdivision |
| **Unmanaged floral richness** | LMM | Gaussian | Property value<br>Property age<br>Property size<br>Forest cover | Subdivision |
| **Managed floral density** | LMM | Gaussian | Property value<br>Property age<br>Property size<br>Forest cover | Subdivision |
| **Managed floral richness** | LMM | Gaussian | Property value<br>Property age<br>Property size<br>Forest cover | Subdivision |
| **Nectar robbery (Y/N)** | Generalized linear mixed effects (GLMM) | Binomial | Property value<br>Property age<br>Property size<br>Forest cover<br>Unmanaged floral density<br>Unmanaged floral richness<br>Managed floral density<br>Managed floral richness<br>Florivory<br>Focal plant flower number | Subdivision |
| **Florivory** | GLMM | Negative binomial | Property value<br>Property age<br>Property size<br>Forest cover<br>Unmanaged floral density<br>Unmanaged floral richness<br>Managed floral density<br>Managed floral richness<br>Flower size (PC1)<br>Focal plant flower number | Subdivision |
| ***Gelsemium* pollen load** | GLMM | Negative binomial | Property value<br>Property age<br>Property size<br>Forest cover<br>Unmanaged floral density<br>Unmanaged floral richness<br>Managed floral density<br>Managed floral richness<br>Nectar robbery<br>Florivory<br>Flower morph (pin/thrum)<br>Focal plant flower number | Subdivision |

(*Continued*)

**Table 1.** (Continued)

| Response | Model type | Error distribution | Fixed effects | Random effect |
|---|---|---|---|---|
| **Non-*Gelsemium* pollen load** | GLMM | Negative binomial | Property value<br>Property age<br>Property size<br>Forest cover<br>Unmanaged floral density<br>Unmanaged floral richness<br>Managed floral density<br>Managed floral richness<br>Nectar robbery<br>Florivory<br>Flower morph (pin/thrum)<br>Focal plant flower number | Subdivision |
| **Fruit set** | LMM | Gaussian | Unmanaged floral density<br>Unmanaged floral richness<br>Nectar robbery<br>Florivory<br>*Gelsemium* pollen load<br>Non-*Gelsemium* pollen load<br>Focal plant flower number | Subdivision |
| **Seeds per fruit** | GLMM | Negative binomial | Unmanaged floral density<br>Unmanaged floral richness[1]<br>Nectar robbery<br>Florivory<br>*Gelsemium* pollen load<br>Non-*Gelsemium* pollen load<br>Fruit herbivory<br>Focal plant flower number | Subdivision |

proportion, of flowers damaged by florivores because a candidate submodel with proportion as the response variable violated model assumptions even after transformation. While fruit herbivory was an interaction of potential interest, it was not a focal interaction in this study and preliminary investigation indicated that it was not influenced by any socioeconomic or vegetation variable (results not shown). Given the limitations imposed by sample size, we omitted it as a response variable in the SEM.

In addition to the vegetation attributes that were included as both predictor and response variables, we included several other ecological variables only as predictors (Table 1). The number of flowers on the focal *Gelsemium* plant was included in our models of nectar robbery, florivory, conspecific and heterospecific pollen load, fruit set, and seeds per fruit. Focal *Gelsemium* flower size (PC1 from the PCA of floral morphology traits; see Floral traits, above) was included as a predictor of florivory. Focal *Gelsemium* flower morph (pin/thrum) was included as a predictor for conspecific and heterspecific pollen load. Finally, fruit herbivory was included as a predictor of seeds per fruit.

Component mixed-effects submodels were formulated using the package 'glmmTMB'[48]. Before formal analysis, we visually checked regression plots for all hypothesized relationships for evidence of strong nonlinearity. Finding none, we continued with the assumption of linear response of all variables to all predictors tested. Data for nectar robbery and florivory came from two censuses, so for those submodels we calculated subdivision-level floral resource metrics as the average across both transects (managed) or all radii (unmanaged) from the floral survey closest to each census date. Because pollen counts and plant reproduction outcomes were not associated with particular censuses, for these submodels we calculated a single value for each floral metric by averaging across censuses.

Analysis of our initial SEM indicated significant departure from expectations of directed separation (Fisher's C = 232.6, p < 0.001). Inspection of model output indicated that this was due to correlation between 1) managed floral richness and density and 2) focal plant flower number and unmanaged floral density. We had no reason to posit a directional causal link between variables in these cases, but correlation between them makes biological sense, so we updated the model to include them as correlated variables.

## Results

Property-scale managed floral richness and density both varied significantly across subdivision (richness: $\chi^2$ = 111.6, df = 25, p < 0.001; density: $\chi^2$ = 218.8, df = 25, p < 0.001). However, within subdivisions, among-property variation in both density and, particularly, richness was greater at many subdivisions than the difference in mean richness or density between the most and least flower-rich subdivisions (density: 9 of 26 subdivisions; richness: 23 of 26 subdivisions; S3 Fig). Similarly, unmanaged floral richness and density, assessed in 10 m-radius surveys around focal *Gelsemium*, differed significantly among subdivisions (richness: $\chi^2$ = 48.2, df = 23, p = 0.002; density: $\chi^2$ = 176.1, df = 23, p < 0.001), though within-subdivision variation exceeded the variation in mean between most and least flower-rich sites for richness at 3 subdivisions and density at 4 subdivisions (S3 Fig).

Nectar robbing occurrence was very low overall this year (<10% of plants experienced robbing, and for plants that were robbed, median proportion of flowers robbed was 0.2) and did not vary significantly across subdivisions ($\chi^2$ = 32.9, df = 27, p = 0.2) or between censuses ($\chi^2$ = 0.27, df = 1, p = 0.6). Florivory, on the other hand, did vary significantly among subdivisions ($\chi^2$ = 89.8, df = 27, p < 0.001; proportion of plants with florivore damage ranged from 0.11 to 0.76 across subdivisions and maximum number of flowers damaged per plant ranged from 1–31) and between censuses ($\chi^2$ = 22.0, df = 1, p < 0.001). Similarly, *Gelsemium* pollen load ($\chi^2$ = 49.7, df = 27, p = 0.005; median ranged from 298–1464 *Gelsemium* pollen grains per stigma across subdivisions), fruit set ($\chi^2$ = 75.4, df = 25, p < 0.001; median 0–1 proportional fruit set), and seeds per fruit ($\chi^2$ = 40.0, df = 25, p = 03; median 0–57 seeds per fruit) all significantly varied among subdivisions.

Our hypothesized SEM provided a reasonable representation of the relationships among variables (Fisher's C = 117.8, df = 116, p = 0.44; p > 0.05 indicates reasonable fit). Twelve of the 71 possible links were supported at p < 0.05, with 11 additional links at 0.05 < p < 0.1, though for most variables we were able to explain only a small proportion of overall variance (Fig 3, Table 2). For example, for the estimates of plant reproduction, we only explained 5% of the variation in fruit set and 20% in seeds per fruit. All measured socioeconomic attributes were associated with some vegetation attribute. Property age was negatively associated with subdivision forest cover (older subdivisions had less forest cover). Property value was positively associated with managed floral density but not richness, and property size was positively associated with unmanaged floral density but not richness. Property value also had direct negative association (i.e., not mediated by vegetation attributes) with two interaction measures, heterospecific pollen load and florivory (though the latter was marginally nonsignificant). Similarly, property age had a relatively strong but marginally nonsignificant positive direct association with nectar robbery. There were no direct links from socioeconomic attributes to *Gelsemium* reproduction.

Among vegetation attributes, managed floral richness had the strongest association with species interactions, being negatively correlated with florivory and both conspecific and heterospecific pollen load. Managed floral density, by contrast, had marginally non-significant positive associations with both conspecific and heterospecific pollen load. Unmanaged floral

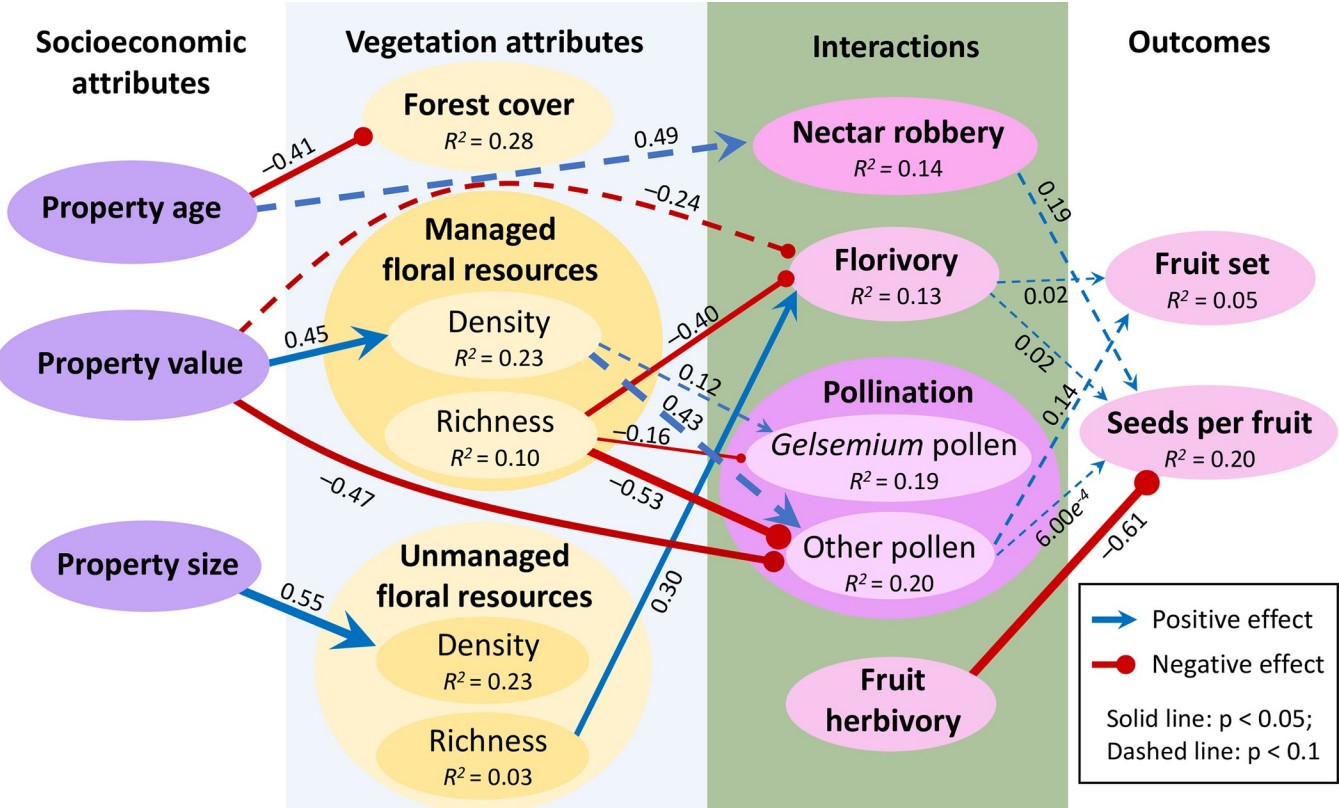

**Fig 3. Results from piecewise SEM analysis.** Only links with statistical support at p < 0.05 (solid lines) and 0.05 < p < 0.1 (dashed line) are included. $R^2$ values indicate marginal $R^2$ for that variable from the full model, including links that are not indicated here (see Table 2 for full model output). Arrowhead indicates a positive association, while filled circle indicates a negative association. Line thickness is proportional to the standardized effect size.

richness was associated with increased florivory, opposite to the relationship with managed floral richness, but showed no association with pollen loads. Neither subdivision-scale forest cover nor unmanaged floral density were associated with any species interaction or *Gelsemium* reproduction measure.

In addition to socioeconomic and vegetation attributes, florivory was strongly positively associated with the number of flowers on the plant (unsurprising given that the response variable was the number of flowers with florivore damage) and pollen load was higher in thrum compared to pin flowers, significantly so for *Gelsemium* pollen and marginally so for non-*Gelsemium* pollen (not included in Fig 3; see Table 2). No vegetation attributes or focal plant traits affected the likelihood of nectar robbing.

Because property value was positively associated with managed floral density, which in turn was marginally associated with heterospecific and conspecific pollen load, property value was indirectly associated with *Gelsemium* pollination. However, in the case of heterospecific pollen, the magnitude of this indirect association was smaller than the direct association with property value (0.19 vs. −0.47); therefore, the net association with property value was negative for heterospecific pollen load and weakly positive for conspecific pollen load (Table 3).

The only significant relationship between species interactions and plant reproduction was a strong negative association between fruit herbivory and seeds per fruit. Other interactions had marginal effects: fruit set and seeds per fruit were both marginally positively associated with florivory and non-*Gelsemium* pollen load. Seeds per fruit, but not fruit set, was marginally

positively associated with nectar robbing. Neither measure of *Gelsemium* reproduction was associated with conspecific pollen load. Because property value, managed floral density and richness, and unmanaged floral richness influenced florivory and/or non-*Gelsemium* pollen load, all these subdivision attributes were indirectly (albeit weakly) associated with fruit set and seeds per fruit (Table 3). Similarly, seeds per fruit was indirectly associated with property age via the association between property age and nectar robbery (Table 3).

**Table 2.** Model output for piecewise SEM relating subdivision socioeconomic and vegetation attributes to *Gelsemium*-insect interactions and *Gelsemium* reproduction.

| Response | Predictor | Standardized Estimate | DF | Crit. value | P |
|---|---|---|---|---|---|
| **Forest cover**<br>Marginal $R^2$: 0.28<br>Conditional $R^2$: NA | Property value | -0.08 | 24 | -0.47 | 0.6 |
| | **Property age** | **-0.41** | **24** | **-2.28** | **0.03\*** |
| | Property size | 0.30 | 24 | 1.70 | 0.1 |
| **Unmanaged floral density**<br>Marginal $R^2$: 0.23<br>Conditional $R^2$: 0.25 | Property value | -0.20 | 19 | -1.35 | 0.2 |
| | Property age | -0.11 | 19 | -0.65 | 0.5 |
| | **Property size** | **0.55** | **19** | **3.43** | **0.003\*\*** |
| | Forest cover | -0.26 | 19 | -1.52 | 0.1 |
| **Unmanaged floral richness**<br>Marginal $R^2$: 0.03<br>Conditional $R^2$: 0.63 | Property value | 0.14 | 19 | 0.71 | 0.5 |
| | Property age | 0.14 | 19 | 0.63 | 0.5 |
| | Property size | 0.07 | 19 | 0.28 | 0.8 |
| | Forest cover | -0.05 | 19 | -0.19 | 0.9 |
| **Managed floral density**<br>Marginal $R^2$: 0.23<br>Conditional $R^2$: 0.44 | **Property value** | **0.45** | **19** | **2.72** | **0.01\*** |
| | Property age | -0.07 | 19 | -0.36 | 0.7 |
| | Property size | -0.16 | 19 | -0.89 | 0.4 |
| | Forest cover | -0.27 | 19 | -1.38 | 0.2 |
| **Managed floral richness**<br>Marginal $R^2$: 0.10<br>Conditional $R^2$: 0.38 | Property value | -0.20 | 19 | -1.12 | 0.3 |
| | Property age | 0.11 | 19 | 0.52 | 0.6 |
| | Property size | -0.07 | 19 | -0.35 | 0.7 |
| | Forest cover | -0.10 | 19 | -0.47 | 0.6 |
| **Nectar robbery**<br>Marginal $R^2$: 0.14<br>Conditional $R^2$: 0.14 | Property value | -0.26 | 277 | -0.83 | 0.4 |
| | Property age | 0.49 | 277 | -1.95 | 0.06• |
| | Property size | 0.15 | 277 | 0.59 | 0.6 |
| | Forest cover | -0.07 | 277 | -0.22 | 0.8 |
| | Unmanaged floral density | -0.36 | 277 | -0.94 | 0.3 |
| | Unmanaged floral richness | -0.01 | 277 | -0.05 | >0.9 |
| | Managed floral density | -0.04 | 277 | -0.10 | >0.9 |
| | Managed floral richness | -0.11 | 277 | -0.38 | 0.7 |
| | Focal plant flower number | 0.22 | 277 | 1.66 | 0.1 |
| **Florivory**<br>Marginal $R^2$: 0.13<br>Conditional $R^2$: 0.13 | Property value | -0.24 | 269 | -1.75 | 0.08• |
| | Property age | -0.02 | 269 | -0.16 | 0.9 |
| | Property size | 0.02 | 269 | 0.14 | 0.9 |
| | Forest cover | 0.16 | 269 | 1.13 | 0.3 |
| | Unmanaged floral density | 0.03 | 269 | 0.24 | 0.8 |
| | **Unmanaged floral richness** | **0.30** | **269** | **2.61** | **0.009\*\*** |
| | Managed floral density | 0.13 | 269 | 0.85 | 0.4 |
| | **Managed floral richness** | **-0.40** | **269** | **-2.92** | **0.004\*\*** |
| | Focal plant flower size (PC1) | 0.15 | 269 | 1.53 | 0.1 |
| | **Focal plant flower number** | **0.64** | **269** | **5.65** | **<0.001\*\*\*** |

*(Continued)*

**Table 2.** (Continued)

| Response | Predictor | Standardized Estimate | DF | Crit. value | P |
|---|---|---|---|---|---|
| *Gelsemium* pollen load<br>Marginal $R^2$: 0.19<br>Conditional $R^2$: 0.22 | Property value[1] | – | – | – | – |
| | Property age[1] | – | – | – | – |
| | Property size[1] | – | – | – | – |
| | Forest cover[1] | – | – | – | – |
| | Unmanaged floral density | 0.00 | 191 | 0.00 | >0.9 |
| | Unmanaged floral richness | 0.03 | 191 | 0.53 | 0.6 |
| | Managed floral density | 0.12 | 191 | 1.80 | 0.07• |
| | **Managed floral richness** | **-0.16** | **191** | **-2.19** | **0.03*** |
| | Nectar robbery (Y/N) | -0.02 | 191 | -0.12 | 0.9 |
| | Florivory | -0.01 | 191 | -0.38 | 0.7 |
| | **Flower morph (pin/thrum)** | | **1** | **23.08** | **<0.001*** |
| | Focal plant flower number | 0.12 | 191 | 1.91 | 0.06• |
| Non-*Gelsemium* pollen load<br>Marginal $R^2$: 0.20<br>Conditional $R^2$: 0.41 | **Property value** | **-0.47** | **191** | **-2.05** | **0.04*** |
| | Property age | 0.20 | 191 | 1.19 | 0.2 |
| | Property size | -0.34 | 191 | -1.40 | 0.2 |
| | Forest cover | 0.25 | 191 | 1.34 | 0.2 |
| | Unmanaged floral density | 0.14 | 191 | 0.64 | 0.5 |
| | Unmanaged floral richness | 0.22 | 191 | 1.47 | 0.1 |
| | Managed floral density | 0.43 | 191 | 1.89 | 0.06• |
| | **Managed floral richness** | **-0.53** | **191** | **-2.95** | **0.003**** |
| | Nectar robbery (Y/N) | -0.38 | 191 | -1.56 | 0.1 |
| | Florivory | -0.04 | 191 | -1.17 | 0.2 |
| | Flower morph (pin/thrum) | | 1 | 2.73 | 0.1 |
| | Focal plant flower number | 0.10 | 191 | 1.04 | 0.3 |
| Fruit set<br>Marginal $R^2$: 0.05<br>Conditional $R^2$: 0.21 | Unmanaged floral density | 0.01 | 187 | 0.41 | 0.7 |
| | Unmanaged floral richness | -0.02 | 187 | -0.60 | 0.5 |
| | Nectar robbery (Y/N) | 0.02 | 187 | 0.45 | 0.7 |
| | Florivory | 0.02 | 187 | 1.96 | 0.07• |
| | *Gelsemium* pollen load | -0.09 | 187 | -1.23 | 0.2 |
| | Non-*Gelsemium* pollen load | 0.14 | 187 | 1.89 | 0.06• |
| | Focal plant flower number | 0.00 | 187 | -0.23 | 0.8 |
| Seeds per fruit<br>Marginal $R^2$: 0.20<br>Conditional $R^2$: NA | Unmanaged floral density | 0.02 | 140 | 0.63 | 0.5 |
| | Unmanaged floral richness | 0.03 | 140 | 0.70 | 0.5 |
| | Nectar robbery (Y/N) | 0.19 | 140 | 1.77 | 0.08• |
| | Florivory | 0.02 | 140 | 1.88 | 0.07• |
| | *Gelsemium* pollen load[1] | – | – | – | – |
| | Non-*Gelsemium* pollen load | 7.00E-04 | 140 | 1.86 | 0.06• |
| | **Fruit herbivory** | **–0.61** | **140** | **–4.64** | **<0.001*** |
| | Focal plant flower number[1] | – | – | – | – |
| **Correlated variables** | | | | | |
| **Managed floral density ~~ Managed floral richness** | | **0.55** | **44** | **4.23** | **<0.001*** |
| **Unmanaged floral density ~~ Focal plant flower number** | | **0.16** | **394** | **3.27** | **<0.001*** |

Boldface indicates a predictor with a significant effect at $p < 0.05$

•$p < 0.1$

*$p < 0.05$

**$p < 0.01$

***$p < 0.001$.

[1]Because of convergence issues, these variables were not included in the submodels used in piecewise SEM analysis. Alternative analysis of the full submodel, including these predictors, using the package 'glmmTMB' eliminated convergence issues, and in all cases the omitted predictors had $p > 0.5$.

**Table 3. Net association between socioecological attributes of subdivisions and ecological interactions and *Gelsemium* reproduction outcomes.** Net association includes both direct and indirect associations; only variables with a significant or marginally significant (p < 0.1) indirect association are included. For net standardized association, the first number includes only significant associations at p < 0.05; the number in parentheses additionally includes associations at p < 0.1.

| Attribute variable | Interaction or outcome variable | Net standardized effect |
|---|---|---|
| Property age | Seeds per fruit | 0 (0.09) |
| Property value | Florivory | 0 (−0.24) |
| Property value | *Gelsemium* pollen | 0 (0.05) |
| Property value | Other pollen | −0.47 (−0.28) |
| Property value | Fruit set | 0 (−0.07) |
| Property value | Seeds per fruit | 0 (−0.0003) |
| Managed floral density | Fruit set | 0 (0.06) |
| Managed floral density | Seeds per fruit | 0 (0.0003) |
| Managed floral richness | Fruit set | 0 (−0.08) |
| Managed floral richness | Seeds per fruit | 0 (−0.008) |
| Unmanaged floral richness | Fruit set | 0 (0.006) |
| Unmanaged floral richness | Seeds per fruit | 0 (0.006) |

## Discussion

The goal of this study was to identify causal pathways linking socioeconomic and ecological attributes of subdivisions to the frequency and outcomes of interactions between *Gelsemium* and its pollinators, nectar robbers, and florivores. We found that socioeconomic attributes influenced both managed and unmanaged floral density but not species richness, while species interactions were more strongly associated with flowering plant richness (both managed and unmanaged). Therefore, direct relationships between socioeconomic attributes, particularly property value, and species interactions were unexpectedly stronger than vegetation-mediated relationships. The associations between species interactions and plant reproduction were small and not statistically significant. The only interaction that strongly affected seeds per fruit was fruit herbivory, an interaction that was not commonly noted in other studies of *Gelsemium* [37, 38]. Ultimately, we were able to explain only a small component of the variation in interaction frequency and plant reproductive outcomes across subdivisions (marginal $R^2$ ranged from 0.05 for fruit set to 0.20 for heterospecific pollen load). Below, we discuss the significant relationships this study uncovered, and why we found relatively little signal of socioecological attributes of subdivisions affecting *Gelsemium*-insect interactions.

It is well established that socioeconomic status is associated with vegetation composition in residential areas, at least in North America [43, 49, 50], and the positive relationship between property value and managed floral density that we found is consistent with this. Higher property values are associated with the wealth of residents, and wealth in turn provides the resources and time to garden and/or to pay others for landscaping services, likely leading to increased ornamental plantings and high floral density. However, property value was not associated with managed floral species richness, indicating a decoupling of floral density and richness, perhaps driven by the high appeal of a consistent suite of ornamental species. This is in contrast to Martin et al. [51], who found that socioeconomic status was associated with plant species richness but not abundance in an arid metropolis, suggesting that climate may mediate the relationship between socioeconomic status and vegetation richness and density [50].

It is interesting that property-level attributes were associated with ecological attributes of subdivisions beyond the boundaries of the properties themselves. For example, we found a positive association between property size and unmanaged floral density in forest fragments.

This highlights how the ecology of seminatural habitat fragments embedded in a matrix of residential development is influenced by the social processes that shape that development. Further research into the mechanisms underlying these links is needed. One fruitful avenue may be to explore the historical development of urbanizing regions. For example, the negative association between property age and subdivision forest cover likely reflects not an effect of age *per se*, but rather changing regulations and development patterns. In the study region, older subdivisions were built primarily closer to the urban core and tend to be higher density, while newer subdivisions tend to be lower density and include substantial acreage left to woodland.

Previous research has found neighborhood effects on both antagonistic and mutualistic species interactions to be highly context- and scale-dependent [32, 52, 53], and our results fit this pattern. While we found correlations between floral resources and species interactions, they were not consistent across interaction type or floral resource measure, and so did not support our hypothesis that both antagonistic and mutualistic interactions would show similar correlations with managed floral resource availability. Managed floral resource density weakly increased stigma pollen loads, but there was a stronger opposing negative relationship between managed floral species richness and both pollen loads and florivory. Unmanaged floral richness had a smaller and unexpectedly positive association with florivory (but no relationship to pollen load). Managed species comprised most floral resources available at the time of *Gelsemium* flowering. The contrasting effects of density and richness suggest that while high density of floral resources may increase insect populations and interaction frequency, increased floral species richness may lead to dilution via a sampling effect [54], whereby species-rich communities are more likely to include species highly attractive to potential *Gelsemium* interactors and therefore reduce interaction frequency. Nevertheless, it is surprising that the negative relationship between managed floral richness and pollen load was substantially stronger for non-*Gelsemium* pollen than for *Gelsemium* pollen; we expected the opposite. The mechanism underlying the negative association between managed species richness and multiple interaction measures deserves further study.

Surprisingly, the only significant association between socioeconomic attributes and interaction measures was the direct negative association between property value and non-*Gelsemium* pollen load; associations mediated by vegetation attributes were only marginally significant and substantially weaker. Though only marginally significant, a similar direct negative association with property value was observed for florivory. Whether these associations reflect conditions directly related to property value (e.g., insecticide use to control pests could be higher in more expensive subdivisions) or covariation between property value and an unmeasured ecological factor is not clear.

We found little effect of the focal *Gelsemium*-insect interactions on measures of female reproductive success. This is consistent with some results from previous studies in this system, which found no strong relationships between nectar robbing or florivory and female reproduction [36, 42, 55]. However, we were surprised to find no effect of conspecific stigma pollen load on either fruit set or seeds per fruit, given previous evidence that *Gelsemium* is pollen-limited [35] and positive relationships between *Gelsemium* pollen receipt and reproduction in urban sites [42]. High within-plant variability in pollen receipt could explain the lack of relationship between conspecific pollen load and reproduction, since we only measured stigma pollen load on one or two flowers per plant. Unexpectedly, the one interaction that did significantly influence *Gelsemium* reproduction was fruit damage by an unknown herbivore. This was not a focal interaction for this study, and preliminary investigation indicated no links between fruit herbivory and either vegetation or socioeconomic attributes, so it was not included as a response variable in our broader SEM. Given its apparent importance in determining seed production in *Gelsemium*, however, the identity and ecology of this fruit

herbivore deserves further study. Of course, while observational studies can generate valuable insight into the processes underlying ecological patterns, they cannot determine causality. Particularly in a study such as this, which considered multiple interaction types and environmental parameters, choices about which variables to measure and which to ignore constrain our inference. As we have noted, it is not possible for us to say from this study whether the observed associations represent direct effects of the predictor variable or covariance with another, unmeasured driver. The patterns noted here can serve to guide future investigation, including manipulative experiments that allow for evaluation of causation.

Overall, subdivisions' socioecological attributes explained little of the variation in *Gelsemium*-insect interactions. There are several possible explanations for this. First, it may be that ecological differences among subdivisions were insufficient to impact interactions. From previous work in this system, we know that interaction frequency and outcomes differ among subdivision-associated forest fragments and intact forest [35, 38, 42]. But it may be that these effects stem from suburban development *per se*, and the specific trajectories taken by different subdivisions have minimal additional effect. Alternatively, the weak links between socioeconomic attributes and interaction frequency or outcome may reflect scale mismatch, with the socioeconomic variables measured at a larger scale than that at which ecological factors influenced *Gelsemium*-insect interactions. Finer-scale analysis of socioeconomic and physical factors is complicated by the resolution of publicly available data. Other methods (e.g., household surveys [26], machine learning combined with remote sensing [17]) could increase the resolution of socioeconomic data, bringing its scale into greater accordance with that of ecological factors, and potentially revealing causal links between socioeconomic factors and ecological outcomes.

One additional goal of this study was to evaluate whether subdivisions could act as units of analysis for understanding how suburbanization shapes ecological interactions. Our results suggest that the answer may be 'no'. While subdivisions differed significantly in both managed and unmanaged floral density and richness, within-subdivision variation in floral density and richness, both among properties for managed floral resources and among survey radii for unmanaged floral resources, was large. Indeed, for most subdivisions, within-site variation eclipsed the range of variation in mean values across all subdivisions. That is, despite relative homogeneity in the built environment and socioeconomic attributes of households within a subdivision, this did not translate into homogeneity in vegetation management–or, we suspect, ecological interactions. This reinforces the need to choose the appropriate scale(s) at which variables are measured, particularly when trying to integrate both social and ecological processes in the same analysis.

## Supporting information

**S1 Table L. ist of subdivisions (suburban residential developments) included in study.**
(DOCX)

**S1 Fig. Violin plots showing distribution of values for subdivision-level socioecological variables included in analysis.**
(TIF)

**S2 Fig. Pearson's correlation matrix for subdivision-level socioecological variables.** Larger circle and darker color indicate a stronger correlation; positive correlations are shown in blue and negative correlations in red. Significant correlations ($p < 0.05$) are indicated with an asterisk.
(TIF)

**S3 Fig.** Variation in vegetation attributes (A-D) and interactions (E-I) within and among subdivisions. Large circles and error bars indicate subdivision-level mean value ± 1 s.e.; smaller points indicate individual observations. In each panel, the x-axis is ordered by mean value of the variable of interest. Note $\log_{10}$ transformation of y-axis in B & D.
(TIF)

## Acknowledgments

Rachel Danford provided key assistance with field project management, data collection, digital data processing, and exploratory analyses. Thanks to the landowners who provided access to field sites, and to the dedicated field crew who assisted with data collection, in particular Becca Walling and Craig See.

## Author Contributions

**Conceptualization:** Lynn S. Adler, Rebecca E. Irwin, Paige S. Warren.

**Data curation:** Gordon Fitch, Lynn S. Adler, Rebecca E. Irwin, Paige S. Warren.

**Formal analysis:** Gordon Fitch.

**Funding acquisition:** Lynn S. Adler, Rebecca E. Irwin, Paige S. Warren.

**Investigation:** Lynn S. Adler, Rebecca E. Irwin, Paige S. Warren.

**Methodology:** Lynn S. Adler, Rebecca E. Irwin, Paige S. Warren.

**Project administration:** Lynn S. Adler, Rebecca E. Irwin, Paige S. Warren.

**Visualization:** Gordon Fitch.

**Writing – original draft:** Gordon Fitch.

**Writing – review & editing:** Gordon Fitch, Lynn S. Adler, Rebecca E. Irwin, Paige S. Warren.

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
