## [Decision Letter · Decision Letter 0]

16 Jul 2024

PONE-D-24-20754Socioecological drivers of mutualistic and antagonistic plant-insect interactions in suburban landscapesPLOS ONE

Dear Dr. Fitch,

Thank you for submitting your manuscript to PLOS ONE. After careful consideration, we feel that it has merit but does not fully meet PLOS ONE’s publication criteria as it currently stands. Therefore, we invite you to submit a revised version of the manuscript that addresses the points raised during the review process.

We look forward to receiving your revised manuscript.

Kind regards,

Andrea Mastinu

Academic Editor

PLOS ONE

Journal Requirements:

2. Thank you for stating the following financial disclosure: "NSF-DEB-0742923 to LSA and NSF-DBI-2109520 to GF"  

3. Please expand the acronym “NSF” (as indicated in your financial disclosure) so that it states the name of your funders in full.

4. Thank you for stating the following in the Acknowledgments Section of your manuscript: "Rachel Danford provided key assistance with field project management, data collection, digital data processing, and exploratory analyses. Thanks to the landowners who provided access to field sites, and to the dedicated field crew who assisted with data collection, in particular Becca Walling and Craig See. This work was funded by the National Science Foundation (NSF-DEB500 0742923 to LSA and DBI-2109520 to GF). Any opinions, findings, and conclusions or recommendations expressed in this material are those of the author(s) and do not necessarily reflect the views of the National Science Foundation."

Please remove any funding-related text from the manuscript and let us know how you would like to update your Funding Statement. Currently, your Funding Statement reads as follows: "NSF-DEB-0742923 to LSA and NSF-DBI-2109520 to GF"

Reviewers' comments:

Reviewer's Responses to Questions

**Comments to the Author**

1. Is the manuscript technically sound, and do the data support the conclusions?

Reviewer #1: Partly

Reviewer #2: Yes

Reviewer #3: Partly

2. Has the statistical analysis been performed appropriately and rigorously? 

Reviewer #1: No

Reviewer #2: Yes

Reviewer #3: Yes

3. Have the authors made all data underlying the findings in their manuscript fully available?

Reviewer #1: Yes

Reviewer #2: Yes

Reviewer #3: Yes

4. Is the manuscript presented in an intelligible fashion and written in standard English?

Reviewer #1: Yes

Reviewer #2: Yes

Reviewer #3: Yes

5. Review Comments to the Author

Reviewer #1: The study addresses an important gap in current research by investigating the influence of socioeconomic and ecological factors on Gelsemium-insect interactions within residential subdivisions. The research methodology is commendable, employing a robust combination of field surveys and statistical analyses to explore complex relationships. Furthermore, the discussion section offers valuable insights into the implications of the findings and suggests promising directions for future research.

One significant area for improvement lies in the clarity of hypothesis testing. While the study presents hypotheses, there is ambiguity in how these hypotheses were tested and addressed in the analysis. Clarifying the hypotheses and their specific relevance to the research objectives would enhance the reader's understanding of the study's purpose and findings.

Moreover, the manuscript would benefit from a more detailed explanation of the results. While the results are presented comprehensively, the interpretation sometimes lacks specificity, making it difficult for readers to fully grasp the significance of the observed patterns. Providing deeper insights into the implications of the results and their relevance to the broader research context would strengthen the manuscript's impact.

Additionally, the manuscript would benefit from a more thorough discussion of methodological limitations. While the study design and methods are generally well-described, there is insufficient acknowledgment and exploration of potential limitations. Discussing the limitations inherent in the study design and data collection processes would enhance the credibility of the findings and help contextualize the conclusions within the broader scientific landscape.

Reviewer #2: a- Species interactions: first of all, this study is not about interactions per se, as there is no data on who visits whom and who eats whom, but it is about the outcome of the (unobserved) interactions, hence about frugivory and pollination: in other words it is about reproductive success. This should be made crystal clear in all parts of the manuscripts, including the title, abstract, aims, hypotheses etc. Results can obviously be interpreted/discussed as a function of changing interactions, but you are not analysing any interaction here! The main flaw of the ms is that it needs to be refocused on what is actually being analysed.

b- Hypotheses. 98-109 You need to explain why you specifically expect the hypotheses, using literature, theory and/or previous exemplary case studies. Otherwise, these you are just listing just ideas as a brainstorming, which is not a very scientific process. Instead, you may appropriately present the hypotheses with the relevant background of literature, especially considering also what stated in my comment “a”.

c- It is not clear in the ms why forest cover should be dependent on property value, age, size. In fact, I think that dependence in the SEM is not justified at all. Forest cover depend on landscape variables mainly and how the urban sprawl was conducted.

42 – Here I would directly introduce that cities are an habitat that is different from the non-urban surroundings, see Biella et al 2022 J Apl Ecol. In turn, different areas of a city might be different micro-habitats based on the variation of biotic and abiotic features within a city (CIT).

47- however, the opposite is much more common actually, with species turnover largely contributing to species interaction change see eg Biella et al 2020 Func Ecol, Carstensen et al 2014 Plos One

71-73 but also subdivisions are the same type of habitat… especially from the methods I understood that you are studying forest fragments, hence again the same type of habitat

91 and elsewhere – it seems a bit out of scope wondering on how vegetation attributes depend on socioeconomic factors: the link between aim 1 and 2 should be better explained. If this paper in on the reproductive success of that specific vine plant, why we are looking at aim1 at all?

220-229 pollination and not pollinators

289 – predictors are crucial, why in supplementary material? Plase put it in the main text

Table 1 – why forest cover is not in the models of Fruit set and Seed set?

Results – why property value and plant richness should decrease the “other” pollen? I imagined the opposite.

Results – where are the results of aim a?

482- and abstract. I have to say that it is not very clear what a subdivision is and why it is relevant as a “special” case, I would ask to explain a bit better the context (above, not here)

Reviewer #3: I read the Ms “Socioecological drivers of mutualistic and antagonistic plant-insect interactions in suburban landscapes” by Fitch et al. I think this Ms includes a large field and detailed work to explore the correlations between socioeconomic attributes (mean property value, mean property size, subdivision age), vegetation attributes (forest cover and richness and density of managed and unmanaged floral resources), species interactions on a focal native species (Gelsemium; in particular, conspecific and heterospecific pollen deposition, nectar robbing, florivory), and the reproductive traits of this focal plant (fruit set, seeds per fruit).

I have some comments and suggestions that could be considered in the new version.

1) In the introduction there are justifications to relate socioeconomic status and biodiversity. The cited references (32-34) are from the USA and I think to be acknowledge because in some other regions of the world, periurbans with people with low incomes manage biodiversity for food and other uses enhancing biodiversity. I am not questioning the Lines 166-168: “Property value is an indicator of wealth and socioeconomic status, which we predicted would be positively associated with the density and diversity of managed floral resources [32,33]. Property age is related to the types of yard plantings and the amount of time they have been in place – both of which can be related to the availability of managed floral resources [32,34]. Lot size is an indicator of the amount of plantable yard space, and therefore the availability of managed floral resources”; I am just saying this perspective is for a particular region and could not be generalized.

2) In the result section, some “effects” and “influences” are described. For example, Lines 375-378 “The only significant relationship between species interactions and plant reproduction was a strong negative effect of fruit herbivory on seeds per fruit. Other interactions had marginal effects: fruit set and seeds per fruit were both marginally positively influenced by florivory and non-Gelsemium pollen load”. Or in Lines 380-384: “Because property value, managed floral density and richness, and unmanaged floral richness influenced florivory and/or non-Gelsemium pollen load, all these subdivision attributes indirectly (albeit weakly) affected fruit set and seeds per fruit (Table 2). Similarly, seeds per fruit was indirectly influenced by property age via its effect on nectar robbery (Table 2)”. I think the experimental design cannot detect effects or influences between variables. There are correlations between variables (mostly very low negative or positive correlations) that are very interesting to be described but cannot be mentioned as effects or influences.

3) In the Discussion, Lines 389-393 (“We found that socioeconomic attributes influenced both managed and unmanaged floral density but not species richness, while flowering plant richness (both managed and unmanaged) had a greater effect on species interactions. Therefore, direct effects of socioeconomic attributes, and particularly property value, on species interactions were unexpectedly stronger than vegetation-mediated effects”) authors mentioned this statements that are neither reflected by the experimental design nor by their results. Moreover, some lines below mentioned somewhat contradictory sentences (Lines 399-401. Below, we discuss the significant relationships this study uncovered, and why we found relatively little signal of socioecological attributes of subdivisions affecting Gelsemium-insect interactions). I think, these paragraphs could be better aligned using better terms or expanding the explanations?.

4) The same comment for the introduction when discussing their findings because most previous studies (including one paper with a meta-analysis) were performed in USA and Europe (Lines 402-404. It is well established that socioeconomic status is associated with vegetation composition in residential areas [32,39,40], and the positive effect of property value on managed floral density that we found is consistent with this). I think this trend is for some societies.

5) If authors want to include a hypothesis (I think it could not be necessary because this is an exploratory study but very useful), it will be rejected (entirely) or accepted as was presented. The formal proposal to present a hypothesis does not admit a “mixed support”. See Lines 427-431. “We hypothesized that landscape-scale floral resource availability would increase interaction frequency by augmenting insect population size, while co-flowering neighbors would dilute visitation. However, support for this hypothesis was mixed. While managed floral resource density weakly increased stigma pollen loads, there was a stronger opposing negative effect of managed floral species richness on both pollen loads and florivory”.

6) Lines 439-452. “Of the socioeconomic attributes we assessed, both property age and property value influenced species interactions…..” I think results do not support this paragraph. Moreover, some lines below (Lines 473-476. “Alternatively, the weak links between socioeconomic attributes and interaction frequency or outcome may reflect scale mismatch, with the socioeconomic variables measured at a larger scale than that at which ecological factors influenced Gelsemium-insect interactions”, text contradicts the paragraph presented above.

Minor comments

Line 152. Correct typo (surveys. census)

6. PLOS authors have the option to publish the peer review history of their article (what does this mean?). If published, this will include your full peer review and any attached files.

Reviewer #1: No

Reviewer #2: No

Reviewer #3: **Yes: **Leonardo Galetto

---

## [Author Response · Author response to Decision Letter 0]

23 Aug 2024

Thanks to the reviewers for their constructive comments. Below, we respond to each comment. Line numbers used in our comments refer to the revised manuscript file with track changes visible.

Reviewer #1: 

The study addresses an important gap in current research by investigating the influence of socioeconomic and ecological factors on Gelsemium-insect interactions within residential subdivisions. The research methodology is commendable, employing a robust combination of field surveys and statistical analyses to explore complex relationships. Furthermore, the discussion section offers valuable insights into the implications of the findings and suggests promising directions for future research.

Thank you for your positive comments!

One significant area for improvement lies in the clarity of hypothesis testing. While the study presents hypotheses, there is ambiguity in how these hypotheses were tested and addressed in the analysis. Clarifying the hypotheses and their specific relevance to the research objectives would enhance the reader's understanding of the study's purpose and findings.

We have substantially re-worked the section of the manuscript outlining our research objectives and hypotheses, including adding more explanation and justification for them. The revised section of the manuscript can be found in L89-126, as well as additions to the Methods section. (See also our response to comments from Reviewer #2.)

Moreover, the manuscript would benefit from a more detailed explanation of the results. While the results are presented comprehensively, the interpretation sometimes lacks specificity, making it difficult for readers to fully grasp the significance of the observed patterns. Providing deeper insights into the implications of the results and their relevance to the broader research context would strengthen the manuscript's impact.

We have revised the Discussion to improve the interpretation of the results, including an expanded discussion of the implications of the patterns we observed. See the track-change document for full details on these updates.

Additionally, the manuscript would benefit from a more thorough discussion of methodological limitations. While the study design and methods are generally well-described, there is insufficient acknowledgment and exploration of potential limitations. Discussing the limitations inherent in the study design and data collection processes would enhance the credibility of the findings and help contextualize the conclusions within the broader scientific landscape.

In line with your comment here and comments from Reviewer #3, we have revised the manuscript throughout to clarify that the observational approach we used limits our ability to draw inference about causal relationships underlying the patterns we observe. We have also added a section to the Discussion that explicitly discusses some of the limitations of the study (L507-515):

Of course, while observational studies can generate valuable insight into the processes underlying ecological pattern, they cannot determine causality. Particularly in a study such as this, which considered multiple interaction types and environmental parameters, choices about which variables to measure and which to ignore constrain our inference. As we have noted, it is not possible for us to say from this study whether the observed associations represent direct effects of the predictor variable or covariance with another, unmeasured driver. The patterns noted here can serve to guide future investigation, including manipulative experiments that allow for evaluation of causation.

--

Reviewer #2: 

a- Species interactions: first of all, this study is not about interactions per se, as there is no data on who visits whom and who eats whom, but it is about the outcome of the (unobserved) interactions, hence about frugivory and pollination: in other words it is about reproductive success. This should be made crystal clear in all parts of the manuscripts, including the title, abstract, aims, hypotheses etc. Results can obviously be interpreted/discussed as a function of changing interactions, but you are not analysing any interaction here! The main flaw of the ms is that it needs to be refocused on what is actually being analysed.

You are certainly correct that we did not measure interactions per se in this study. But we think it is incomplete to say that we measured only reproductive success. Instead, we measured both reproductive success (via fruit set and seed production) and proxies for interaction frequency (pollen receipt and levels of damage from herbivores, florivores, and nectar robbers). It is common in ecological studies of species interactions to assess proxies of the interaction rather than the interaction itself, and still make inferences about interactions (e.g., for pollination: Tur et al. 2016, Ecology Letters [stigma pollen loads]; for herbivory: O’Connor 2009, Ecology [plant biomass]). In some cases, measuring this proxy is in fact more informative than measuring the interaction itself. For example, knowing how much of a flower was consumed by florivores is, we would argue, more important to understanding the florivore-plant interaction than knowing how many times a florivore was recorded on the flower. That is, an observation of an interaction occurring is not evidence of its ecological significance – particularly since, for many plant-insect interactions, it can be hard to distinguish between an ecologically trivial interaction (e.g., a pollinator lands on a flower) and a significant one (e.g., a pollinator consumes nectar resources from a flower and in the process transfers pollen to that flower). For these reasons, we think it is reasonable and appropriate to use the term “interactions” to describe the metrics we use here.

That said, we have re-read the manuscript with this comment in mind. To clarify exactly what we measured in this study, we have made the following changes:

L 99-102: As proxies for antagonistic interactions, we assessed the proportion of flowers with evidence of nectar robbery and/or florivory, and the proportion of fruits with herbivore damage; for mutualistic interactions, used pollen receipt as a proxy for pollination.

(previous wording: For antagonistic interactions, we assessed the frequency of nectar robbing and floral and fruit herbivory;…)

Caption for Figure 2 (in part):

Focal plants were censused for evidence of florivory, nectar robbing, and pollination (estimated as pollen deposition), and were assessed for fruit and seed production. 

(previous wording: Focal plants were monitored for interactions with florivores, nectar robbers, and pollinators (estimated as pollen deposition), and were assessed for fruit and seed production.)

b- Hypotheses. 98-109 You need to explain why you specifically expect the hypotheses, using literature, theory and/or previous exemplary case studies. Otherwise, these you are just listing just ideas as a brainstorming, which is not a very scientific process. Instead, you may appropriately present the hypotheses with the relevant background of literature, especially considering also what stated in my comment “a”.

We have revised this section to include further explanation and justification for the hypothesized relationships, including citations to relevant literature for each prediction. The revised passage (L102-127) now reads:

We hypothesized that socioeconomic attributes would influence interactions both directly (e.g., by influencing management intensity [26]) and indirectly via their effects on vegetation attributes [27,28], though we did not have a priori expectations for the direction of these effects (Fig 1). All the insect species involved in the interactions we assessed are dietary generalists, so we predicted that vegetation attributes such as high managed floral density and richness would increase populations of antagonists and pollinators, and would therefore increase interaction frequency [29,30]. At the same time, greater floral resources can decrease interaction frequency via a dilution effect [31,32]. We predicted that dilution would dominate at local scales (i.e., in the 20m radius around focal plants), leading to negative correlations between unmanaged floral resources and interaction metrics, but that the net effect of landscape-scale floral resource availability would be increased interaction frequency, and therefore a positive association between managed floral resources and interaction metrics (Fig 1). We expected conspecific and heterospecific pollen receipt to be strongly correlated, but predicted that increased non-Gelsemium floral density and richness would lead to more pollinator switching and therefore increase heterospecific pollen receipt [33,34]. We expected socioeconomic and vegetation attributes to influence reproductive outcomes for Gelsemium only via effects on the frequency of species interactions (rather than direct effects), and that increased interactions with antagonists and heterospecific pollen would reduce reproductive success whereas increased conspecific pollen receipt would increase Gelsemium reproduction, given previous evidence for pollen limitation [35]. By integrating social and ecological drivers of plant-insect interactions into a single causal network, we hope to increase mechanistic understanding of the influence of urban development on species interactions. 

c- It is not clear in the ms why forest cover should be dependent on property value, age, size. In fact, I think that dependence in the SEM is not justified at all. Forest cover depend on landscape variables mainly and how the urban sprawl was conducted.

Forest cover in this case is a measure of forest cover within the subdivision. Therefore, we hypothesized that it would be related to property value, age, and lot size since all these factors are influenced by the subdivision development process. We have explained this with the following addition to the “Socioeconomic and land cover attributes – data sources” section of the Methods (L189-192): 

We predicted that forest cover would be related to socioeconomic attributes of subdivisions because choices made during the development process (e.g., house size, lot size) would simultaneously influence property value and remaining forest cover, while age since development would be related to post-development forest regrowth. 

See also our discussion of the negative association between subdivision age and forest cover in the Discussion (L449-454): 

For example, the negative association between property age and subdivision forest cover likely reflects not an effect of age per se, but rather changing regulations and development patterns. In the study region, older subdivisions were built primarily closer to the urban core and tend to be higher density, while newer subdivisions tend to be lower density and include substantial acreage left to woodland.

42 – Here I would directly introduce that cities are an habitat that is different from the non-urban surroundings, see Biella et al 2022 J Apl Ecol. In turn, different areas of a city might be different micro-habitats based on the variation of biotic and abiotic features within a city (CIT).

Thank you for the suggestion. We have added the following sentence where you indicated: 

As a result of these changes, urban environments support biological communities that are often distinct from adjacent non-urban counterparts [e.g., 5,6].

47- however, the opposite is much more common actually, with species turnover largely contributing to species interaction change see eg Biella et al 2020 Func Ecol, Carstensen et al 2014 Plos One

We respectfully disagree with this assessment. In our opinion, there is not a clear consensus on the relative importance of species turnover vs. interaction rewiring in determining differences in interaction networks across environmental gradients. For example, in the Carstensen et al. paper you reference, interaction turnover among species shared across sites was of an equal magnitude to species turnover. At any rate, the point of this statement was not to say that interaction turnover is more important than (or even as important as) species turnover in determining network structure, but rather to point out that it has been less thoroughly studied. We stand by this statement, and contend that it is in part an assumption that rewiring without species loss is less significant than species turnover that has led to the relative neglect of the phenomenon.

71-73 but also subdivisions are the same type of habitat… especially from the methods I understood that you are studying forest fragments, hence again the same type of habitat

Subdivisions are the same type of habitat in the same way that cities are the same type of habitat – that is, their similarity can encompass a great deal of variation. The argument we are making here is that within a subdivision, variation in the built environment and associated socioecological attributes is less than in a comparable area of non-master-planned urban environment. This reduction in fine-scale heterogeneity could allow for better inference about the specific socioecological attributes that are driving patterns of association between the urban environment and ecological interactions. This point was made in the lines directly preceding the ones referenced by the reviewer’s comment, and we have not modified it.

91 and elsewhere – it seems a bit out of scope wondering on how vegetation attributes depend on socioeconomic factors: the link between aim 1 and 2 should be better explained. If this paper in on the reproductive success of that specific vine plant, why we are looking at aim1 at all?

We did not intend for this to be read as stating two aims of the paper, but rather to summarize the structure of the model. We see how the previous wording gives the impression of two aims, and so have re-worded the sentence as follows: 

We used piecewise structural equation modeling to investigate the links among socioeconomic attributes, vegetation attributes, and the frequency and plant reproductive outcome of antagonistic and mutualistic plant-insect interactions.

To your broader question: with this paper, we are looking at how socioeconomic attributes of suburban areas influence interaction frequency and outcomes. As we detail in the introduction and discussion, there are reasonably well-established links between socioeconomic attributes and managed vegetation in urban residential areas, so we believe assessing these links in our study system is essential for a full understanding of the ways that socioeconomics may influence species interactions.

220-229 pollination and not pollinators

We are uncertain what this comment refers to. The lines referenced detail our methods for assessing nectar robbing. Throughout, we have made efforts to clarify, along the lines of your general comment a, that we assessed stigma pollen load, not pollinator visitation, as our measure of pollination.

289 – predictors are crucial, why in supplementary material? Plase put it in the main text

We have moved Table S2 to the main text, where it is now Table 1.

Table 1 – why forest cover is not in the models of Fruit set and Seed set?

Forest cover was not included because we did not hypothesize a relationship between these variables. Forest cover refers to subdivision-scale extent of forested land, and we saw no reason why this would relate to fruit or seed set. If there was a significant correlation between forest cover and either of these variables, this would have been detected in our test of directed separation, but we found no evidence for these relationships (p>0.3 in both cases).

Results – why property value and plant richness should decrease the “other” pollen? I imagined the opposite.

We were also surprised by the negative association between managed plant richness and heterospecific pollen. While the precise mechanism behind this association requires further investigation, we propose the following explanation in our Discussion:

The contrasting effects of density and richness suggest that while high density of floral resources may increase insect populations and interaction frequency, increased fl

---

## [Decision Letter · Decision Letter 1]

10 Sep 2024

PONE-D-24-20754R1Socioecological drivers of mutualistic and antagonistic plant-insect interactions and interaction outcomes in suburban landscapesPLOS ONE

Dear Dr. Fitch,

Thank you for submitting your manuscript to PLOS ONE. After careful consideration, we feel that it has merit but does not fully meet PLOS ONE’s publication criteria as it currently stands. Therefore, we invite you to submit a revised version of the manuscript that addresses the points raised during the review process.

We look forward to receiving your revised manuscript.

Kind regards,

Andrea Mastinu

Academic Editor

PLOS ONE

Journal Requirements:

Reviewers' comments:

Reviewer's Responses to Questions

**Comments to the Author**

1. If the authors have adequately addressed your comments raised in a previous round of review and you feel that this manuscript is now acceptable for publication, you may indicate that here to bypass the “Comments to the Author” section, enter your conflict of interest statement in the “Confidential to Editor” section, and submit your "Accept" recommendation.

Reviewer #3: All comments have been addressed

2. Is the manuscript technically sound, and do the data support the conclusions?

Reviewer #3: Partly

3. Has the statistical analysis been performed appropriately and rigorously? 

Reviewer #3: Yes

4. Have the authors made all data underlying the findings in their manuscript fully available?

Reviewer #3: Yes

5. Is the manuscript presented in an intelligible fashion and written in standard English?

Reviewer #3: Yes

6. Review Comments to the Author

Reviewer #3: I concur with the updates made in the new edition, except for one aspect.

In science, there are two principal types of hypotheses: (a) direct (inductive), which involves searching for patterns without clear expectations, looking for links, correlations, etc., with the anticipation of discovering distinct patterns post-study, which I believe applies here. The authors stated in the Introduction, "We hypothesized that socioeconomic attributes would influence interactions both directly (e.g., by influencing management intensity [26]) and indirectly through their effects on vegetation attributes." The phrase "would influence" implies no need for trend expectations (i.e., predictions). Conversely, the second type of hypothesis, (b) indirect (deductive), necessitates a solid theoretical framework to underpin the expectations (predictions are mandatory).

In the previous round, I stated (with additional clarifications): 5) If authors wish to include a hypothesis—which may not be necessary in an exploratory study but can be beneficial (thus, an indirect type)—it will either be entirely rejected or accepted as presented. This is because including predictions categorizes it as an indirect type. The formal proposition of a hypothesis (when it is contructed as the indirect type) does not allow for "mixed support."

The indirect type requires full support, while the direct type may accept "mixed support" due to the unpredictable patterns that may emerge from the dataset. This observation aligns precisely with the comment from reviewer #1: “One significant area for improvement lies in the clarity of hypothesis testing. While the study presents hypotheses, there is ambiguity in how these hypotheses were tested and addressed in the analysis. Clarifying the hypotheses and their specific relevance to the research objectives would enhance the reader's understanding of the study's purpose and findings”. Your response for this comment (rev # 1) was: “We have substantially re-worked the section of the manuscript outlining our research objectives and hypotheses, including adding more explanation and justification for them”, and for my comment (rev # 3) was: “We think that in this context it is appropriate to talk about “mixed support” for our hypothesis: we predicted that both antagonistic and mutualistic interactions would show similar correlations with managed floral resource availability, but found that the significance and sign of this correlation instead depended on the metric of floral resources used (i.e., density vs. richness) as well as the interaction being assessed. That is, we found some support for our prediction (increased floral resource density was correlated with increased pollen loads, our proxy for interaction with pollinators), but also some opposing responses (floral richness was negatively correlated with pollen load and florivory)”.

I believe there is a misunderstanding regarding the two types of hypotheses. In your paper, the results and discussion should align with the direct hypothesis introduced earlier, as acknowledged in your responses to the three reviewers. It is unnecessary to "force" the inclusion of predictions if they are not required for the study.

In summary, the rationale of the study must be consistent across all sections of the manuscript, including the Introduction, Materials and Methods, Results, and Discussion.

7. PLOS authors have the option to publish the peer review history of their article (what does this mean?). If published, this will include your full peer review and any attached files.

Reviewer #3: **Yes: **Leonardo Galetto

---

## [Author Response · Author response to Decision Letter 1]

13 Sep 2024

Thanks to Reviewer #3 for their constructive comment. Our response and revision are below.

Reviewer #3: I concur with the updates made in the new edition, except for one aspect.

In science, there are two principal types of hypotheses: (a) direct (inductive), which involves searching for patterns without clear expectations, looking for links, correlations, etc., with the anticipation of discovering distinct patterns post-study, which I believe applies here. The authors stated in the Introduction, "We hypothesized that socioeconomic attributes would influence interactions both directly (e.g., by influencing management intensity [26]) and indirectly through their effects on vegetation attributes." The phrase "would influence" implies no need for trend expectations (i.e., predictions). Conversely, the second type of hypothesis, (b) indirect (deductive), necessitates a solid theoretical framework to underpin the expectations (predictions are mandatory).

In the previous round, I stated (with additional clarifications): 5) If authors wish to include a hypothesis—which may not be necessary in an exploratory study but can be beneficial (thus, an indirect type)—it will either be entirely rejected or accepted as presented. This is because including predictions categorizes it as an indirect type. The formal proposition of a hypothesis (when it is contructed as the indirect type) does not allow for "mixed support."

The indirect type requires full support, while the direct type may accept "mixed support" due to the unpredictable patterns that may emerge from the dataset. This observation aligns precisely with the comment from reviewer #1: “One significant area for improvement lies in the clarity of hypothesis testing. While the study presents hypotheses, there is ambiguity in how these hypotheses were tested and addressed in the analysis. Clarifying the hypotheses and their specific relevance to the research objectives would enhance the reader's understanding of the study's purpose and findings”. Your response for this comment (rev # 1) was: “We have substantially re-worked the section of the manuscript outlining our research objectives and hypotheses, including adding more explanation and justification for them”, and for my comment (rev # 3) was: “We think that in this context it is appropriate to talk about “mixed support” for our hypothesis: we predicted that both antagonistic and mutualistic interactions would show similar correlations with managed floral resource availability, but found that the significance and sign of this correlation instead depended on the metric of floral resources used (i.e., density vs. richness) as well as the interaction being assessed. That is, we found some support for our prediction (increased floral resource density was correlated with increased pollen loads, our proxy for interaction with pollinators), but also some opposing responses (floral richness was negatively correlated with pollen load and florivory)”.

I believe there is a misunderstanding regarding the two types of hypotheses. In your paper, the results and discussion should align with the direct hypothesis introduced earlier, as acknowledged in your responses to the three reviewers. It is unnecessary to "force" the inclusion of predictions if they are not required for the study.

In summary, the rationale of the study must be consistent across all sections of the manuscript, including the Introduction, Materials and Methods, Results, and Discussion.

RESPONSE: To be honest, this distinction in the type of support allowed for direct vs. indirect hypotheses is a new concept to us (with a combined ~100 years of experience among co-authors of conducting hypothesis-testing-oriented research). Thank you for bringing it to our attention.

In this case, we think it is worthwhile to elucidate some more specific predicted relationships, in part to assist the reader in navigating what is otherwise a complicated dataset. Indeed, one response to our initial manuscript draft from both Reviewer #1 and Reviewer #2 was that more explication of specific hypotheses would improve the manuscript.

Therefore, we have opted to retain the specific hypothesized relationships in the manuscript. In recognition of the fact that, therefore, our hypothesis must be either completely accepted or rejected, we have re-written the section of the Discussion where we review our hypothesis as follows (L442-449, change italicized):

Previous research has found neighborhood effects on both antagonistic and mutualistic species interactions to be highly context- and scale-dependent [32,52,53], and our results fit this pattern. While we found correlations between floral resources and species interactions, they were not consistent across interaction type or floral resource measure, and so did not support our hypothesis that both antagonistic and mutualistic interactions would show similar correlations with managed floral resource availability. Managed floral resource density weakly increased stigma pollen loads, but there was a stronger opposing negative relationship between managed floral species richness and both pollen loads and florivory.

---

## [Decision Letter · Decision Letter 2]

1 Oct 2024

Socioecological drivers of mutualistic and antagonistic plant-insect interactions and interaction outcomes in suburban landscapes

PONE-D-24-20754R2

Dear Dr. Fitch,

We’re pleased to inform you that your manuscript has been judged scientifically suitable for publication and will be formally accepted for publication once it meets all outstanding technical requirements.

Kind regards,

Andrea Mastinu

Academic Editor

PLOS ONE

Additional Editor Comments (optional):

Reviewers' comments:

Reviewer's Responses to Questions

**Comments to the Author**

1. If the authors have adequately addressed your comments raised in a previous round of review and you feel that this manuscript is now acceptable for publication, you may indicate that here to bypass the “Comments to the Author” section, enter your conflict of interest statement in the “Confidential to Editor” section, and submit your "Accept" recommendation.

Reviewer #3: All comments have been addressed

2. Is the manuscript technically sound, and do the data support the conclusions?

Reviewer #3: Yes

3. Has the statistical analysis been performed appropriately and rigorously? 

Reviewer #3: Yes

4. Have the authors made all data underlying the findings in their manuscript fully available?

Reviewer #3: Yes

5. Is the manuscript presented in an intelligible fashion and written in standard English?

Reviewer #3: Yes

6. Review Comments to the Author

Reviewer #3: The response to the final comment has been received. I consider it to be acceptable as it is in this most recent version.

7. PLOS authors have the option to publish the peer review history of their article (what does this mean?). If published, this will include your full peer review and any attached files.

Reviewer #3: **Yes: **Leonardo Galetto

---

## [Editor Report · Acceptance letter]

10 Oct 2024

PONE-D-24-20754R2 

PLOS ONE

Dear Dr. Fitch, 

I'm pleased to inform you that your manuscript has been deemed suitable for publication in PLOS ONE. Congratulations! Your manuscript is now being handed over to our production team.

Kind regards, 

on behalf of

Dr. Andrea Mastinu 

Academic Editor

PLOS ONE